# Evolution of male life histories and age-dependent sexual signals under female choice

Joel J. Adamson

Ecology, Evolution and Organismic Biology, University of North Carolina, Chapel Hill, NC, United States

Corresponding author
Joel J. Adamson,
adamsonj@ninthfloor.org

## ABSTRACT

Sexual selection theory models evolution of sexual signals and preferences using simple life histories. However, life-history models predict that males benefit from increasing sexual investment approaching old age, producing age-dependent sexual traits. Age-dependent traits require time and energy to grow, and will not fully mature before individuals enter mating competition. Early evolutionary stages pose several problems for these traits. Age-dependent traits suffer from strong viability selection and gain little benefit from mate choice when rare. Few males will grow large traits, and they will rarely encounter choosy females. The evolutionary origins of age-dependent traits therefore remain unclear. I used numerical simulations to analyze evolution of preferences, condition (viability) and traits in an age-structured population. Traits in the model depended on age and condition ("good genes") in a population with no genetic drift. I asked (1) if age-dependent indicator traits and their preferences can originate depending on the strength of selection and the size of the trait; (2) which mode of development (age-dependent versus age-independent) eventually predominates when both modes occur in the population; and (3) if age-independent traits can invade a population with age-dependent traits. Age-dependent traits evolve under weaker selection and at smaller sizes than age-independent traits. This result held in isolation and when the types co-occur. Evolution of age-independent traits depends only on trait size, whereas evolution of age-dependent traits depends on both strength of selection and growth rate. Invasion of age-independence into populations with established traits followed a similar pattern with age-dependence predominating at small trait sizes. I suggest that reduced adult mortality facilitates sexual selection by favoring the evolution of age-dependent sexual signals under weak selection.

## INTRODUCTION

Sexual selection theory studies the exaggeration of secondary sexual traits and corresponding preferences in the opposite sex. The most well-developed portion of this theory describes maintenance of female preferences under indirect (genetic) benefits (*Jones & Ratterman, 2009*). The theory of indicator traits explains the maintenance of female

preferences with genetic correlations between male viability and female preferences (*Kokko, Jennions & Brooks, 2006*). Testing good genes theory requires attention to life-history variables (*Kokko, 2001*). Experimenters often measure correlations between health of sires and survival of offspring (*Evans, Gustafsson & Sheldon, 2011*; *Jacob et al., 2007*; *Jacob et al., 2010*; *Kokko et al., 2002*). Secondary sexual traits, particularly in large vertebrates, require time to grow and young males can enter mating competition against older males with larger weapons (*Pemberton et al., 2004*). Older males provide superior genetic benefits in some cases (*Brooks & Kemp, 2001*). Age-dependent traits occur frequently in nature and form frequent subjects for laboratory studies of sexual selection and coercion (*Bonduriansky & Brassil, 2005*). However, most sexual selection models do not account for the growth of traits and include only simplified life-histories (*Kokko, Jennions & Brooks, 2006*; *Kokko, 2001*; *Kokko et al., 2002*).

Life-history theory suggests that sexual selection theory could benefit from modeling more complex life-histories. Low adult mortality leads to a stable strategy of age-dependent male reproductive effort. *Kokko (1997)* found evolutionarily stable strategies for age-dependent strategies under fairly broad conditions. *Proulx, Day & Rowe (2002)* found that males benefit from increasing mating effort as they age and reproductive opportunities decline. They predicted that condition should be positively correlated with delays in investment. High condition males signal more at older ages. A third, more recent study employing similar techniques found that optimal higher-quality males will postpone trait growth until the onset of breeding (*Rands, Evans & Johnstone, 2011*).

Life-history models and long-term studies of vertebrates suggest a specific class of secondary sexual traits that require explanation. I define age-dependent traits as "quantitative traits" that males grow throughout their reproductive lifetimes. Antlers of deer (*Kodric-Brown & Brown, 1984*), horns on sheep (*Pemberton et al., 2004*; *Coltman et al., 2002*) and body size in pinnipeds (*Clinton & Le Boeuf, 1993*) and primates (*Courtiol et al., 2012*; *Geary, 2002*; *Mace, 2000*) form good examples of age-dependent morphological traits. Others examples come from behavioral traits that change over the lifetime due to experience or aging, such as song repertoires (*Hiebert, Stoddard & Arcese, 1989*; *Gil, Cobb & Slater, 2001*), nest building (*Evans, 1997*), performance ability (*Judge, 2011*; *Verburgt, Ferreira & Ferguson, 2011*; *Ballentine, 2009*; *Garamszegi et al., 2007*), and social connectivity (*Oh & Badyaev, 2010*; *McDonald & Potts, 1994*). I define age-independent traits as morphological patterns that are relatively stable over the lifetime, and qualitatively different from juvenile morphology at the time of breeding. Readers could refer to age-independent traits as "qualitative traits." Plumage patterns in birds (e.g., "breeding plumage") provide examples of age-independent traits. Some plumage patterns do change over the reproductive lifespan (*Evans, Gustafsson & Sheldon, 2011*) making them age-dependent by the definition above.

The evolutionary origins of these age-dependent sexual signals and their corresponding preferences remain unclear, despite the clarity of strategic models. First, when traits require a large investment of resources to grow, they require time to mature. Such a trait would require reduced adult mortality. A strongly age-dependent trait could suffer selection

before it grows enough to make it attractive. Second, frequency dependence crucially affects the origin of costly sexual signals (*Kirkpatrick, 1982*). If we suppose that both trait and preference arise by mutation, then males growing age-dependent traits will rarely encounter choosy females during early evolutionary stages. Sexual selection will have limited opportunity to increase the trait. Such a trait could be eliminated by selection, or lost to drift in finite populations. This introduces a critical time period the trait must survive before proceeding to fixation. Third, signal honesty poses another problem. For some kinds of traits, males will have similar trait values at young ages regardless of variation in condition. Age-dependence of traits thereby weakens both the heritability of the trait and the phenotypic correlation between the trait and condition. Finally, at the evolutionary origin of a trait, only a subset of strategies occurs in the population. Stability of a particular strategy does not tell us which will predominate under such a restricted strategy set. This is a classic problem in game theory, with a voluminous literature (see *Hammerstein, 1998*, and references therein).

A model of evolutionary dynamics would balance the theory of age-dependent traits. I used numerical simulations to investigate the evolution of an age-dependent trait under sexual selection. All males start with similar trait values and grow their traits in a condition-dependent manner. The model population consists of haploid, dioecious (male and female) individuals. Selection on the male trait produces the age structure of the population. For the purposes of the model, age-dependent and age-independent traits differ in that age-independent traits are fully grown at the time of first breeding, and only vary between males due to condition-dependence. I ask three specific questions. First, how does the strength of selection and the growth rate of the trait affect its evolution? Second, I investigate which mode of trait development (age-dependent versus age-independent) will eventually predominate when both are initially present and rare. Third I ask if age-independence can invade a population with established age-dependent traits and preferences. I investigate these questions by examining the differences between conditions for evolution of age-dependent and age-independent traits. I show that strongly age-dependent traits can increase in frequency at smaller sizes than age-independent traits, both in the presence and absence of age-independent traits. Age-dependent traits in my model require weak selection to eventually predominate. My results suggest that weak selection, strong age-dependence and reduced adult mortality can lead to the exaggeration of sexual signals in species with extended lifespans. This suggests age-dependent traits are compatible with life-histories seen in long-lived vertebrates.

## MODEL

My model extends a classic model of female choice (*Kirkpatrick, 1982*) by adding age-dependence, iteroparity, overlapping generations of males, and condition-dependence. The model uses a haploid life cycle (Fig. 1): new zygotes arise from meiosis and undergo viability selection followed by mating. Males that survive the first round of selection and mating proceed to another round of selection and mating, and so on. Males provide no direct benefits to females, and can mate with multiple females. Each female mates once and

**Figure 1 Haploid life cycle for cohorts of males: haploid males emerge, grow, undergo selection, then mating, then repeated episodes of selection and mating.** Meiosis then produces new zygotes following mating. Mutation in condition loci occurs before selection.

lives for one episode of viability selection followed by mating. I assume a large population size so that we can ignore the effects of genetic drift. All male mortality results from selection on the trait and condition phenotypes except for males in the terminal age class, who are removed from the simulation unconditionally (see Eq. (10)). Similarly, females do not suffer costs of choice, and all selection on females reflects selection on condition.

The model genome consists of five diallelic loci: two "condition" or "intrinsic viability" loci (C), the trait locus (T), the preference locus (P) and the age-dependent mode of expression locus (F). I refer to alleles of interest, e.g., the preference or trait allele, or beneficial condition alleles, with the subscript 2, and denote their frequencies by lower-case $p$ with a subscript corresponding to the locus. For example $p_P$ represents the frequency of the choosiness allele $P_2$. The alleles at the condition loci are either "beneficial" or "deleterious." The number of beneficial alleles across loci adds up to an individual's condition phenotype C. Biased mutation from beneficial to deleterious ($C_2 \rightarrow C_1$) occurs in condition loci at a rate of 0.001 per individual per generation; mutation occurs in the zygote stage, before the first round of selection (see Fig. 1).

Males carrying the $T_1$ allele do not produce the trait, regardless of condition. Males produce the trait if they have the trait allele $T_2$. A male aged $y$ with condition phenotype C carrying $T_2$ has trait size

$$t(C, y) = be^{Cy} \tag{1}$$

where $b$ is a parameter controlling the size of the trait, that I refer to as the "growth coefficient." I chose this exponential function to emphasize three characteristics: (1) all males display the same trait size at age 0, as long as they carry the trait allele; (2) a large disparity in size between young and old males; and (3) simple scaling of the male trait size via the growth coefficient ($b$). Other trait functions do occur in nature (see *Johnson & Hixon, 2011*; *Poissant et al., 2008*), and could have different consequences for evolutionary dynamics (see Discussion). Figure 2 shows that age-specific trait size linearly depends on the growth coefficient (the constant $b$). The growth coefficient linearly corresponds to the size of the trait across constant age and condition. A larger value of $b$ in a particular population (i.e., simulation) signifies that males of a particular genotype attain larger trait values than they would in populations with smaller $b$-values.

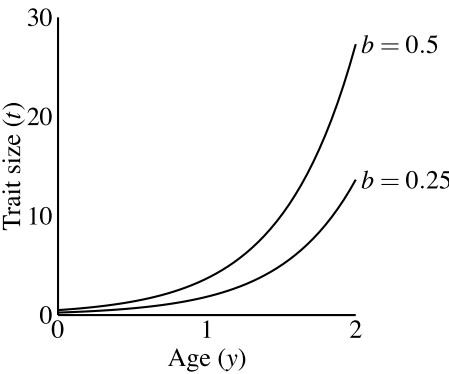

**Figure 2 Male trait growth for high-condition males at two values of the growth coefficient, $b = 1.0$ and $b = 0.5$.** Trait size at age 0 equals $b$.

The P locus controls mate choice behavior of females. Mate choice occurs by relative preference (as in *Kirkpatrick, 1982*) as a function of trait size. Females carrying $P_2$ encountering a male with trait size $t$ are $\phi(t) = 1 + \alpha t$ times more likely to mate than non-choosy females. Non-choosy females, carrying $P_1$, mate randomly ($\phi(t) = 1$). The mating process normalizes female mating frequency such that all females have equal mating success (see Eq. (5)). In other words, all females mate in each mating cycle. Choosy females do not suffer any viability or opportunity costs.

The F locus controls mode of development. Males carrying the $F_1$ allele show age-dependent expression, whereas carriers of the $F_2$ allele express the trait throughout their lives at one of three levels in a particular simulation: (1) $t(C,0) = b$, the trait value of a 0-year-old; (2) $t(C, y_{max})$, the trait value of the oldest males in the population (still dependent on condition); or (3) $\bar{t}$, the population mean trait value. The third set of simulations sought to create a population where age-independent ($F_2$) males were of intermediate attractiveness, between young or unornamented males and older, age-dependent males. Fixation of age-dependence in this case shows that significant mating advantages accrue later in life, despite a period of lesser attractiveness early in life. Fixation of age-independence, on the other hand, would show that early-life attractiveness and costs were more important than potential gains from mating later in life. Contrast this with the second set of simulations: age-dependent males only reach the attractiveness of their age-independent counterparts, when they reach the final age class ($y_{max}$). Age classes run from 0 (youngest) to $y_{max}$. A youngest age class of 0 conveniently yields young males the trait size of $b$ in Eq. (1), as well as using the same indexing convention as the computer simulation. The number of age classes in the population is $y_{max} + 1$.

For the third set of simulations, where $F_2$ males carried $t = \bar{t}$, I updated $\bar{t}$ in every iteration. My goal was ensuring that a class of males of intermediate attractiveness persisted in the population. Therefore all males (regardless of condition) carrying $F_2$ received $\bar{t}$ as their trait value for a particular episode of mating, then I updated their values in the next iteration, following changes in $\bar{t}$. Furthermore, the value of $\bar{t}$ used in these simulations reflected the full range of variation in condition, despite the lack of condition-dependence

for $F_2$ males. I therefore calculated the mean trait as

$$\bar{t} = \frac{\sum_{t=0}^{t_{max}} t f(t, y)}{y_{max} + 1} \tag{2}$$

where $f(t, y)$ describes the frequency of males with trait value $t$ at age $y$ over $y_{max} + 1$ age classes. The average is taken over all males, from unornamented males to the maximum trait size of $t_{max} = b e^{C y_{max}}$, where $C$ represents the largest possible number of condition alleles (i.e., number of condition loci). Males carrying $F_2$ contributed to the population mean as if their traits were age-dependent, i.e., contributing $t(C, y) = b e^{Cy}$ to the calculation in Eq. (2). If the trait allele ($T_2$) were to spread, then a contribution of $\bar{t}$ by $F_2$ males would depress the trait value for $F_2$ males and produce a delay in following phenotypic changes. Although these procedures reduced biological realism from an individual perspective, they maintained the population genetic conditions relevant to the question at hand.

I used one level of expression per simulation, including simulations where the male population was fixed for $F_2$ (an age-independent population). In a different set of simulations $F_2$ initially occurred at a low frequency, comparable to the frequency of the trait, so that age-independent and age-dependent expression were in competition.

Summed Gaussian functions of trait and condition describe viability for males and females:

$$w_m(C, t) = 1 + \exp\left(-\frac{(C - C)^2}{2\mu^2}\right) + \exp\left(-\frac{t^2}{2\nu^2}\right) \tag{3a}$$

$$w_f(C) = 1 + \exp\left(-\frac{(C - C)^2}{2\mu^2}\right) \tag{3b}$$

where males have $C$ condition loci, $\mu$ sets the relative strength of selection for condition, $\nu$ determines the relative strength of selection against the trait. Smaller values of $\mu$ and $\nu$ correspond to stronger selection (see Fig. 3).

The frequency of haplotypes changes through viability selection:

$$P_i'(y) = \frac{P_i(y) W_i(y)}{\bar{W}(y)} \tag{4}$$

where $P_i(y)$ and $P_i'(y)$ represent the frequency of haplotype $i$ at age $y$ before and after selection, respectively. $W_i(y)$ correspondingly represents the viability of haplotype $i$ at age $y$ and $\bar{W}(y)$ represents mean viability within age class $y$. The matrix $\mathbf{M}$ expresses the probability of mating between a female of genotype $i$ and a male of genotype $j$:

$$\mathbf{M}_{ij} = \frac{P_i' \sum_{y=0}^{y_{max}} \phi_i\left(t_j(y)\right) P_j'(y) \pi(y)}{\sum_l \sum_{\upsilon=0}^{y_{max}} \phi_i(t_l(\upsilon)) P_l'(\upsilon) \pi(\upsilon)} \tag{5}$$

where $\pi(y)$ represents the frequency of males of age $y$, and $t_j(y)$ represents the trait size of a male of genotype $j$ at age $y$. Condition is not an argument to the $t()$ function in this case, as

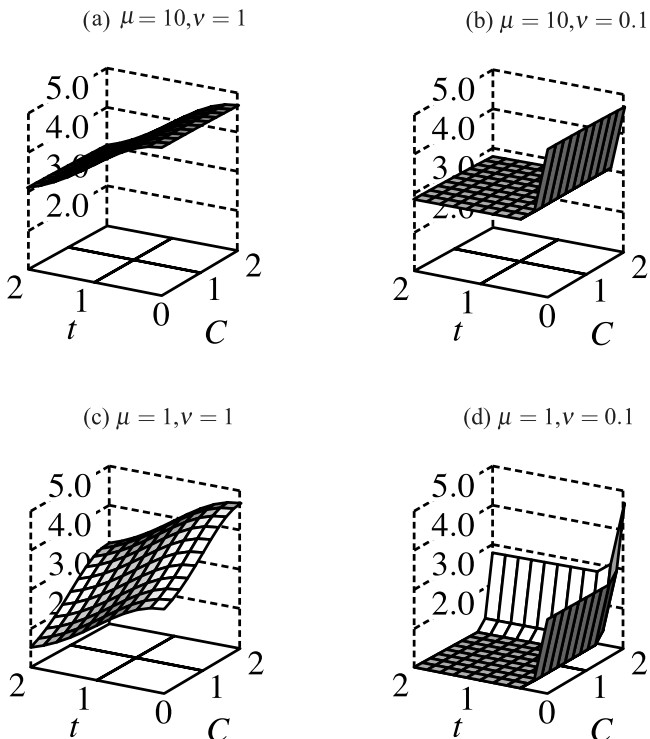

**Figure 3** **Fitness surfaces for males bearing ornaments: fitness slopes away from its maximum at $t = 0$ as the trait grows, and increases with more condition alleles.** Decreasing $\mu$ decreases fitness for males with fewer than 2 condition alleles. When selection on condition weakens ($\mu = 10$, (A) and (B)) the fitness profile flattens in the $C$-dimension; compare this to $\mu = 1$ ((C) and (D)) where males with high-viability alleles at both condition loci have significantly higher fitness than those with 1 or none. Direct selection on the trait ($\nu$) follows a similar profile except that $\nu \leq 1$ such that only trait-less males have significantly higher fitness than trait-bearing males in all simulations.

it was in Eq. (1) since the genotype $j$ specifies the male's condition. Equation (5) expresses the frequency of matings by summing over male ages the product of female mating rate ($\phi$, a function of trait size) and the frequency of the male haplotype $j$ in the general population, i.e., adjusted for the age structure $\pi$. The subscript $i$ on mating rate refers to choosy versus non-choosy females. The denominator adds up the age-dependent sums across all male haplotypes. The ratio of these summed terms then yields the probability that a male breeding with a female of haplotype $i$ is of haplotype $j$. Multiplying by the female haplotype $i$ frequency after selection $P'_i$ yields the frequency of $ij$ pairs after mate choice.

Summing the product of mating probabilities and recombination probabilities across **M** yields the frequency of new zygotes

$$P'_i(0) = \sum_{jk} R_{jk \to i} M_{jk} \tag{6}$$

where $R_{jk \to i}$ represents the proportion of $jk$ matings that yield genotype $i$ after recombination (*Bürger, 2000*). Condition loci recombine freely ($r = 0.5$) with each other and other loci; other loci recombine at arbitrary frequencies ($0 \leq r \leq 0.5$; see Table 1).

**Table 1** List of variables and parameters with typical values.

| Symbol | Meaning | Values |
|---|---|---|
| $t$ | Male trait size | 0–27.30 |
| $\alpha$ | Coefficient of preference | 0.6, 1 |
| $\mu$ | Coefficient of selection on condition | 1, 10 |
| $\nu$ | Coefficient of selection on trait | 0.0001–1.0 in steps of 0.02 |
| $r$ | Recombination fraction | 0.25, 0.5 |
| $\lambda$ | Geometric rate of increase (initial age structure) | 1.0 |
| $b$ | Coefficient of trait growth | 0.0–0.5 in steps of 0.02 |
| $\mathcal{C}$ | Number of condition loci | 2 |
| $y_{max}$ | Oldest age for males | 2 |
| $y_{max} + 1$ | Number of age classes | 3 |

Condition loci recombine freely so that they will represent unlinked loci far away in the genome, and could also represent multiple unlinked loci (*Rowe & Houle, 1996*). The trait and preference loci recombine at arbitrary frequency since prior works show that recombination frequency affects indirect selection on preference (*Kirkpatrick & Barton, 1997*; *Kirkpatrick, 1982*).

The relative size of the zygote class is found by the age-weighted sum of the mean fecundities of all adult age classes:

$$\pi'(0) = \sum_{y=0}^{y_{max}} \bar{m}(y)\pi(y) \tag{7}$$

where $m(y)$ is the fecundity of an individual aged $y$. The new relative size of an adult age class is given by the mean viability of the age class:

$$\pi'(y) = \bar{W}(y)\pi(y) \tag{8}$$

$$= \pi(y)\sum_i P_i(y)W_i(y). \tag{9}$$

I then calculate the new age distribution by dividing by the sum of all new age class sizes (*Moorad & Promislow, 2008*):

$$\pi''(y) = \frac{\pi'(y)}{\sum_{\upsilon=0}^{y_{max}}\pi'(\upsilon)}. \tag{10}$$

I calculated the initial age structure by specifying $\lambda$, the geometric rate of increase for a population in stable age distribution. I then used this Gaussian survivorship function centered at 0 to calculate survival probabilities:

$$l(y) = \exp\left(\frac{-y^2}{2}\right). \tag{11}$$

**Peer**J

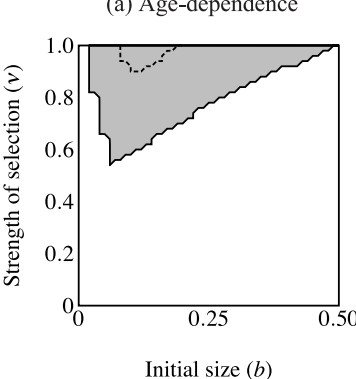
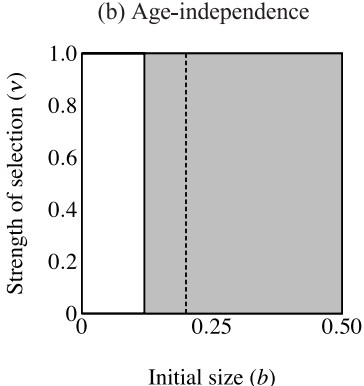

**Figure 4 Region of trait fixation (light gray) for age-dependent (A) and age-independent traits (B) in a plane defined by strength of selection against the trait ($v$) and the growth coefficient of the trait ($b$).** Dashed line indicates region of fixation under $\alpha = 0.6$; solid lines indicates region of fixation for $\alpha = 1.0$. The region of fixation for $\alpha = 0.6$ is contained within the region of fixation for $\alpha = 1.0$ in both panels. Under age-dependence $b$ is the value of the trait for a 0-year old male. The $b$-axis corresponds linearly to the trait size of ornamented males ($t(C, y) = be^{Cy}$), such that populations depicted further to the right on the $b$-axis will have larger average trait sizes. Smaller values of $v$ correspond to stronger selection.

The age distribution is then given by (see *Charlesworth, 1994*):

$$\pi(y) = \frac{\lambda^{-y}l(y)}{\sum_{v=0}^{y_{max}}\lambda^{-v}l(v)}. \tag{12}$$

All simulations started with the trait allele $T_2$ and preference allele $P_2$ at non-zero frequencies in the youngest age-class, and zero in the older age-classes. Simulations ran until: (1) the trait allele $T_2$ fixed; (2) the trait allele $T_2$ or preference allele $P_2$ was lost (frequency dropped below $10^{-12}$); (3) the preference allele $P_2$ fixed; (4) the Euclidean distance between successive generations in preference and trait allele frequencies dropped below $10^{-9}$; or (5) the simulation ran for 10 million iterations.

## RESULTS

### Selection on male trait and growth

The first set of simulations sought to determine the important parameters for evolution of the age-dependent trait compared with an age-independent trait. The age-independent trait was the same size as the eventual size of the age-dependent trait. Age-dependent simulations ran with coefficient of preference ($\alpha$), recombination frequency ($r$) and selection on condition ($\mu$) at the values indicated in Table 1. I used initial values of $p_C = 0.01$, $p_P = 0.1$, $p_T = 0.001$, and $p_F = 0.0$. I compare this to a population where all initial values were the same except that males were fixed for $F_2$, i.e., male trait expression was age-independent and expressed at the largest size attainable ($t = b\exp(Cy_{max})$). I analyzed the relative roles of selection intensity and trait size by plotting the equilibrium value of $p_T$ (fixation versus loss of the trait allele) over a plane defined by $0.00001 \leq v \leq 1.0$ and $0 \leq b \leq 0.5$ (see Table 1; see Figs. 4 and 6).

The area of parameter space where the trait fixes depends on three parameters: $\alpha$ (strength of preference), $b$ ("growth coefficient"; see Eq. (1)) and $\nu$ (strength of selection; Fig. 4). Recombination frequency ($r$) and selection on condition ($\mu$) do not appear to qualitatively affect the results in these simulations. Strength of preference affects the "readiness to mate" of females over the range of trait values present in these simulations: when $\alpha = 0.6$ a choosy female is 9.20 times more likely to mate with a 2 year-old, high-condition male with a growth coefficient of $b = 0.25$. At the higher value of $\alpha = 1.0$, a choosy female is 14.65 times more likely to mate with the same trait-bearing male. We see a qualitative difference between populations with age-dependent traits and populations with age-independent traits at both values of $\alpha$. The selection parameter $\nu$ determines the pattern of fixation for age-dependent traits, but has no effect on age-independent traits. At $\alpha = 0.6$ the trait fixes in a very small portion of the $b$–$\nu$ plane near $\nu = 1.0$ in age-dependent simulations. The traits fixes above the threshold $b$ value of 0.20 in the corresponding age-independent simulations. At $\alpha = 1.0$ the regions of fixation are larger, but the qualitative differences between age-dependent and age-independent expression remain: the age-dependent trait fixes in a roughly triangular region characterized by relatively weak selection and containing a region of small initial trait sizes (Fig. 4A). This contrasts to age-independent simulations, where again above a threshold size ($b = 0.12$) the trait fixes independently of selection intensity (Fig. 4B).

Figure 5 shows a sample of linkage disequilibrium trajectories from the four regions defined by fixation and loss of age-dependent and age-independent traits. Each point on a curve represents the statistical association (linkage disequilibrium) between a beneficial condition allele and the ornament allele. A positive association indicates that the trait allele occurs in the same genotype with a beneficial condition allele more often than expected by chance. A negative association indicates that the two alleles are found less often in one genotype than expected by chance. When the alleles randomly associate, e.g., at the beginning of the simulation, or when one or both alleles are fixed or lost, the linkage disequilibrium equals zero. Condition alleles do not fix in these simulations due to biased mutation.

These associations indicate the effect of selection on genotypes that act as indicators of condition, i.e., have both a condition allele and a trait allele. When selection favors condition (always) and does not favor the trait, it will drive alleles apart so that they are rarely found in the same genotype, leading to the negative associations found in Fig. 5. I caution the reader against interpreting the values in Fig. 5 as directly indicative of the strength of overall selection, since the quantities do not include adjustment for age structure. Each point reflects the action of selection on a particular cohort, and not the strength of selection against the trait in the population. For example, the larger absolute values in older age-classes in the figure does not indicate that selection acts more strongly on individuals in that age class. Selection works quite weakly on older age classes, since old individuals are quite rare.

Using these trajectories gives us a slightly clearer picture of the forces acting to generate the pattern of fixation and loss seen in Fig. 4. Each panel shows curves for each of three age

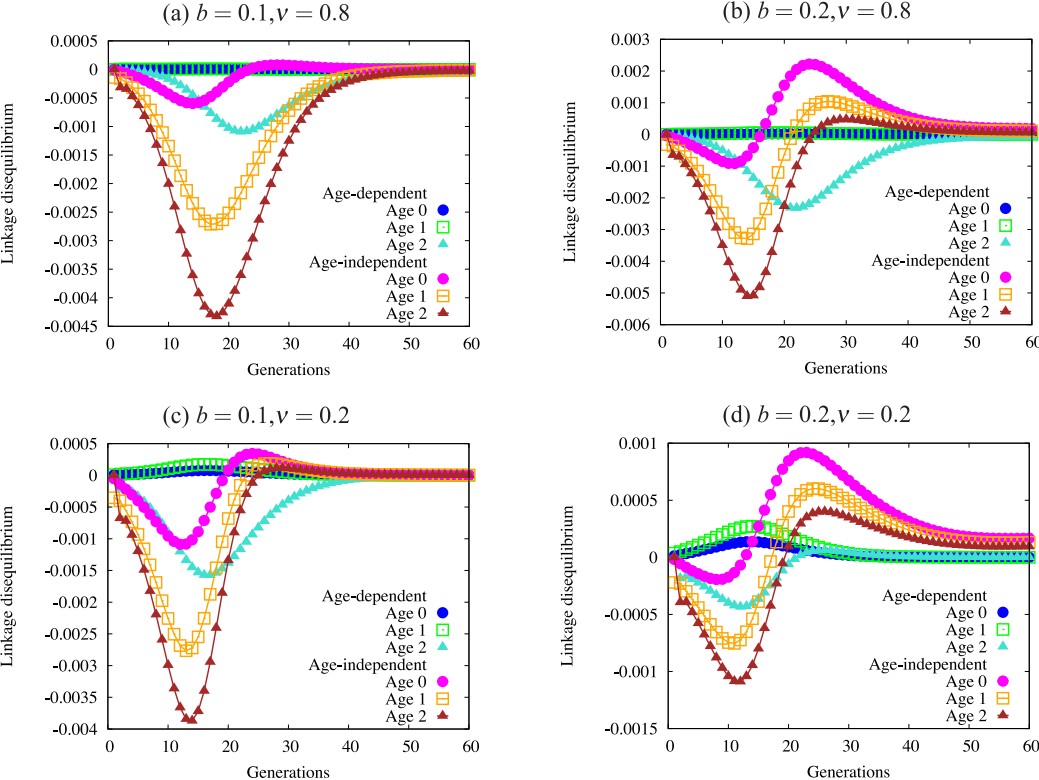

**Figure 5 Trajectories of linkage disequilibrium between the trait and one of the condition loci, with $\alpha = 1.0$.** This quantity reflects selection on the genotype bearing both the trait allele and the beneficial condition allele. The four panels are defined by regions of Fig. 4 with different patterns of fixation: (A) fixation under age-dependence, not under age-independence; (B) fixation under both modes of expression; (C) no fixation under age-dependence or age-independence; and (D) fixation only under age-independence. Note the differing scales on the vertical axis of each panel. Quantities not adjusted for age structure.

classes under both age-dependent and age-independent simulations. All four panels show stronger associations in age-independent simulations than in age-dependent simulations. We can also see that magnitude (absolute value) increases from age 0 to age 2. Increasing negative magnitude reflects that high-condition males, likely to survive to older age classes, are less likely to be ornamented. However, this effect is generally much weaker in age-dependent populations than in age-independent populations (Fig. 5A). Males that reach the oldest age class under age-dependence are more likely to be high-condition and ornamented than under age-independence. More raw material for sexual selection remains within a cohort under age-dependence.

The right-hand column of Fig. 5 shows trajectories for a larger trait ($b = 0.2$). A similar curve appears for age-dependent traits in Fig. 5B as in Fig. 5A. However, the age-independent trait responds to selection rapidly, showing a different shape from age-dependent populations (Figs. 5C and 5D). Readers should keep in mind when viewing this figure that the age structure of the population, especially under strong selection, biases heavily toward young males. Although linkage disequilibrium increases in older age classes,
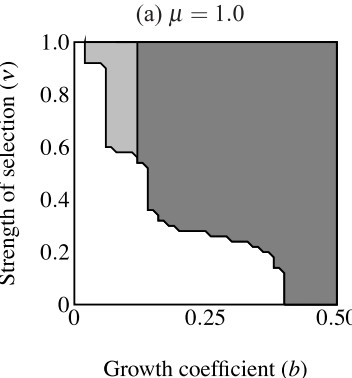
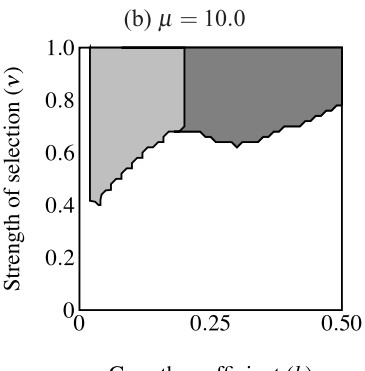

**Figure 6 Regions of fixation for the trait only (light gray) and the trait along with age-independent mode of development allele ($F_2$; dark gray).** Panel (A) shows strong selection on condition and panel (B) shows weak selection on condition. The trait is lost in the white region. $F_2$ does not change from its initial frequency of 0.1 in the light gray region, resulting in a predominance of age-dependent signaling at small trait sizes.

the associations for age class 0 most closely reflect the overall linkage disequilibrium in the population as a whole.

## Mode of development

Another set of simulations sought to determine the crucial parameters favoring age-dependent expression over age-independent expression. These simulations began with polymorphism at the F locus, i.e., $F_2$ began at a low, non-zero frequency. Fixation of $F_2$ depended on the level of trait expression chosen for the age-independent males in that simulation, one of (1) $b$; (2) $\bar{t}$; or (3) $t_{max} = b\exp(Cy_{max})$. I simulated these conditions with initial values of $p_C = 0.01$, $p_P = 0.1$, $p_T = 0.1$, and $p_F = 0.1$ in all age classes. Age-independent males occurred with one particular trait function ($t = b$, $t = \bar{t}$, or $t = b\exp(Cy_{max})$) for a specific simulation: when the $F_2$ allele caused males to have smaller traits than the oldest age-dependent males then $F_2$ was lost everywhere in the $b$–$v$-plane.

When males carrying $F_2$ expressed the trait range of the oldest age-dependent males throughout their lives ($t = b\exp(Cy_{max})$) the $F_2$ allele fixed in an area including the highest intensity of selection (Fig. 6). However, the $F_2$ allele also failed to increase in an area of small initial trait values where the trait did fix. In other words, expression remained polymorphic in this region, with the majority of males displaying age-dependence (light gray in Fig. 6A). This effect is even stronger when selection on condition is relaxed ($\mu = 10$; see Fig. 6B). The area of trait fixation where F remained polymorphic is larger when $\mu = 10$ and includes regions of more intense selection against the trait as well as requiring larger trait sizes for fixation of $F_2$. Age-dependence occurs at larger trait sizes and under more intense selection when the population displays more variance in condition. This suggests that for a given $b$, age-independence only brings males a mating advantage with strong selection on condition.

I also repeated the above simulations with $p_F = 0.9$ initially. For all parameter values $T_2$ was lost below a certain threshold value and above this threshold, both $T_2$ and $F_2$ fixed.

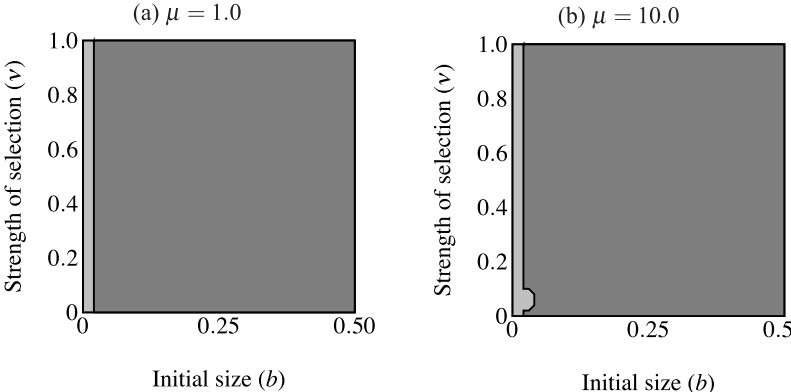

**Figure 7** **Regions of fixation for the age-dependent trait (light gray) and the age-independent trait ($F_2$; dark gray) with the trait and preference initially fixed.** Panel (A) shows strong selection on condition and panel (B) shows weak selection on condition. The $F_2$ allele supplants age-dependent signaling at any strength of selection above a certain threshold. Age-dependent signaling persists at small trait sizes.

This reinforces the above results: when the majority of males display age-independent expression, trait size alone determines the fixation of the trait.

Another set of simulations considered the fixation of the $F_2$ allele with established traits ($p_T = 1.0$) and preferences ($p_P = 1.0$; Fig. 7). The $F_2$ allele was initially rare ($p_F = 0.1$). Individuals carrying $F_2$ displayed the maximum trait size of age-independent males of comparable condition at their maximum age ($t = b \exp(Cy_{max})$). Age-independent signaling came to predominate regardless of selection strength above a certain small size threshold. Below this threshold we can say that age-dependent signaling persisted. The same pattern was observed with a minor exception under strong selection against the trait, regardless of selection on condition (Fig. 7B): selection on condition does not appear to qualitatively affect the size at which age-independent signaling out-competes age-dependent signaling.

## DISCUSSION

Although age-dependent sexual traits are common in nature, their evolution presents several dynamical difficulties. Every sexually selected trait balances the benefits derived through mate choice with the costs of growing a trait. Age-dependent traits undergo a period when selection can easily eliminate them. I have shown that an age-dependent trait can evolve under relatively weak selection at a wider range of trait sizes than can an age-independent trait (Fig. 4). While an age-dependent trait can evolve by producing small traits at young ages when selection is intense, larger initial trait sizes can only occur under proportionally less intense selection. This contrasts strongly with the age-independent simulations. Size of the trait determines the eventual fixation of the age-independent trait allele, regardless of selection intensity. Simulations with genetic variation for mode of development (age-dependent vs. age-independent) show that age-independence only evolves at larger trait sizes and age-dependence can predominate under weaker selection at smaller trait sizes. Invasion of age-independent traits occurred at relatively smaller trait sizes than when increasing from rarity. Again this only depended on trait size. Evolution

of age-independence generally depends on trait size and not on the intensity of selection against the trait.

Altogether the simulations show that small traits in young males "solve" the dynamical problem of reduced heritability. The results suggest that three factors favor the evolution of age-dependent sexual signals: (1) low adult male mortality; (2) weak selection against the trait; and (3) strong age-dependence, i.e., small initial trait sizes. This result makes sense from a life-history strategy perspective. When adult mortality is lower, young males will invest less in reproductive strategies, producing a negative correlation between expected future reproduction and age-specific investment. Males can afford to invest in mate attraction when adult mortality is low. Age-dependence effectively partitions the life-history of males into two stages dominated by two different fitness components. Viability takes precedence at young ages and mating success becomes more important later in life. This result parallels recent work on an abundant primate showing that juvenile survival followed by mating success form the most crucial fitness components (*Courtiol et al., 2012*).

Trajectories of linkage disequilibrium support the above theory (Fig. 5). Selection finds its greatest power early in the life cycle, dealing with the majority of genetic variation in the largest age class. When the additive component of that variation lies hidden within age-dependent traits, it can pass through to older age classes where it will be fully expressed. As males age they reveal additive variance in condition, thus making them more reliable signalers (as in *Proulx, Day & Rowe, 2002*). The trajectories illustrate how selection can only eliminate the age-dependent trait when either selection acts strongly or when the young-age trait is large enough. Weak selection (the top row of Fig. 5) works even weaker on age-dependent populations, so as to barely change allele frequencies until males get older. Age-dependence weakens selection against the trait.

The failure of the age-independent trait to evolve at lower trait sizes (see Fig. 4B) also supports the theory presented above. Using an individual perspective can aid understanding of this result. Consider a population early in the evolution of the trait. The trait and preference occur rarely, and hence viability differences between ornamented and unornamented males dominate the selection differential. Mating success will not form a significant part of fitness due to positive frequency dependence (i.e., rarity of choosy females). From an individual perspective, any trait-bearing male will have lower fitness than an unornamented male. For the trait to be advantageous, it must be large enough for males to gain high mating success despite the rarity of choosy females. On the other hand, for trait-bearing males to avoid viability selection, it must be small. Compare young age-independent males to young age-dependent males in this scenario. Age-independent males that avoid selection never become attractive. Age-dependent males that avoid viability selection, by contrast, do become more attractive. Now consider males bearing large traits. Age-independent males that start out with sufficiently large traits immediately attract choosy females, even though they probably die in the next round of selection. If the reader picks a single $b$-value in Fig. 4, this represent two potential males: (1) a fully-grown age-dependent male and (2) a young age-independent male. Both males display the same

trait size and have the same viability costs. However, the age-dependent male has already lived through two mating episodes and three episodes of selection. After mating both males will die, but the age-dependent male has higher lifetime mating success. Carrying attractive traits at a young age reduces future mating opportunities. We see a threshold of attractiveness in Fig. 4B where age-independent males show large enough traits to avoid this cost. *Kokko (2001)* has already noted that sexual advertisements are life-history traits. Considering the costs and benefits of advertisements requires evaluation of the entire life cycle and all the organism's interactions and requirements (*Badyaev & Qvarnström, 2002*).

The results here support the strategic modeling literature of age-dependent signals (*Rands, Evans & Johnstone, 2011*; *Proulx, Day & Rowe, 2002*; *Kokko, 1997*). *Proulx, Day & Rowe* modeled the situation where male longevity and reproductive opportunities increase — e.g., under a low adult mortality environment — and found that high-condition males downplay their signaling relative to lower condition males, preserving resources for survival. *Kokko (1997)* came to the similar conclusion that young males of lower condition should signal more than their higher-condition cohorts, thus obscuring the observed relationship between genetic quality and trait value. Both studies find that (optimally) males of a given condition signal inversely proportional to number of remaining reproductive attempts (a predictor of condition in any age-class). Selection then favors females that prefer to mate with older males, since they are more likely to be of high condition. These studies model competing strategies, whereas my study uses a condition-dependent and age-dependent trait function to model variation. When selection weakens enough, with a particular developmental trajectory, age-dependent signaling and female preferences evolve in a population genetic model. The evolutionary dynamics, in this case, do mirror the conclusions of the optimization models. These results suggest that with the needed life-history conditions and genetic variation we can expect selection on life-histories to produce age-dependent signaling.

Empirical evidence and my results suggest that extending the male lifespan facilitates sexual selection. When traits are age-dependent they can develop their most exaggerated forms at older ages when selection is less intense. Age-dependent traits or mating success occur in a wide variety of taxa, including mammals (*Poissant et al., 2008*; *Pemberton et al., 2004*; *Clinton & Le Boeuf, 1993*), birds (*Hawkins, Hill & Mercadante, 2012*; *Evans, Gustafsson & Sheldon, 2011*; *Taff et al., 2011*; *Ballentine, 2009*; *Garamszegi et al., 2007*; *Evans, 1997*), fish (*Johnson & Hixon, 2011*; *Jacob et al., 2007*; *Miller & Brooks, 2005*; *Candolin, 2000a*; *Candolin, 2000b*) and insects (*Verburgt, Ferreira & Ferguson, 2011*; *Judge, 2011*; *Kivleniece et al., 2010*; *Jones & Elgar, 2004*; *Jones, Balmford & Quinnell, 2000*). Despite the demographic problems of age-dependence, widespread occurrence of age-dependent traits suggests that life-histories promoting age-dependence are common. Body size or traits directly correlated with body size (e.g., weapons and ornaments) form the most obvious example of age-dependent traits, and should satisfy the assumptions of my model. Researchers found age-dependent sexual selection based on body size in Rocky Mountain Bighorn Sheep (*Ovis canadensis*; *Coltman et al. 2002*), who show a typical life-history characterized by weakening viability selection and increasing heritability

over the lifespan. Older males pay less of a survival cost for larger bodies and larger horn sizes, facilitating greater success in mating competition. Certain behavioral and social traits should display age-dependence under weak natural selection, such as social network connectivity (*McDonald & Potts, 1994*) and song repertoire (*Gil, Cobb & Slater, 2001*). Age-based honesty also creates an effective constraint producing age-dependence. High-condition males cannot bypass age-dependence by "faking" the trait. Certain "skills" such as nest-building (*Evans, 1997*) show this form of honesty.

Readers should consider some limitations of the model I used here. My model inadequately portrays situations where direct costs of female choice impact sexual selection. Direct costs of choice could considerably impact female survival and therefore a full analysis of the life-history implications of sexual selection needs to include a model of the female life-history. Indirect costs, such as increasing frequency of germ-line mutations with male age also greatly affect female choice evolution in long-lived organisms (*Beck & Promislow, 2007*; *Hansen & Price, 1999*). Female choice also depends on female condition, which could produce strong negative linkage disequilibrium (i.e., negative correlations) between condition loci and preference loci under selection against the trait. Condition-dependent female choice could therefore broaden the range of parameters where an age-dependent trait can evolve. The trait function I used here also has somewhat narrow applicability, as it strongly favors old males: age-dependent traits in nature probably favor intermediate-aged males, who have become attractive but have not deteriorated considerably (*Brooks & Kemp, 2001*). A more physiologically realistic trait function would peak in middle-age (*Johnson & Hixon, 2011*), but would tell us little more about the evolution of the preference without considering costs of mating with old (versus intermediate-aged) males. I chose the small maximum age of 2 for computational efficiency. Three age classes also yielded young, middle- and old-aged males without producing old males with enormously exaggerated traits. When males had more age classes, old-age traits became very large and far too advantageous to analyze the tension between sexual selection and viability selection. Males in nature live in more complex age-structured populations with more than three age-classes. My model also uses an effectively infinite population size and overestimates the effect of old males in the population. The effective population size of old age-classes could diminish or fluctuate and drift could eliminate alleles for indicator traits.

Age-dependent signaling offers a testable hypothesis relating life-histories and sexual selection. Researchers should check independent developments of iteroparity and reduced adult mortality for association with age-dependent sexual signals. The weakening of selection associated with lifespan development should facilitate sexual selection by concomitant reductions in selection against outrageous traits at older ages. Further modeling should ask whether senescence and accumulation of mutations could weaken this prediction.

## ACKNOWLEDGEMENTS

I would like to thank: Maria Servedio, Joel Kingsolver, Troy Day, Karin Pfennig, and Haven Wiley for scrutinizing my results; Sumit Dhole, Alicia Frame, Caitlin Stern,

Justin Yeh, Artur Romanchuk and Daniel Promislow for providing helpful discussion and comments; Richard M. Stallman and the GNU Project for guidance with software; the developers of Bazaar, Trac, Guile, GNUPLOT and the GNU C Compiler for creating reliable freedom-respecting software. Samuel Tazzyman, an anonymous reviewer and the editors of PeerJ provided helpful comments that substantially improved the conceptual clarity of this manuscript.

### Funding

This research was supported by NSF DEB-0614166 and NSF DEB-0919018 to Maria Servedio, Ph.D. advisor to the author; these grants were to provide support for graduate students. The funders had no role in study design, data collection and analysis, decision to publish, or preparation of the manuscript.

### Grant Disclosures

The following grant information was disclosed by the author:
NSF: DEB-0614166, DEB-0919018.

### Competing Interests

The author declares that there are no competing interests.

### Author Contributions

- Joel J. Adamson conceived and designed the experiments, performed the experiments, analyzed the data, wrote the paper, computer simulations, analysis of results, created figures.

### Data Deposition

The following information was supplied regarding the deposition of related data:
   Simulation code and data on figshare
   http://figshare.com/articles/Evolution_of_Age_Dependent_Traits/783068
   Evolution of Age-Dependent Traits. Joel Adamson. figshare. http://dx.doi.org/10.6084/m9.figshare.783068
   Retrieved 19:06, Aug 28, 2013 (GMT).

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
