# Peer review of "Evolution of male life histories and age-dependent sexual signals under female choice"

_PeerJ, doi:10.7717/peerj.225_

## Round 0.1 · original submission · Major Revisions

Please make a revised version considering all the comments by the reviewers and resubmit it to PeerJ. Looking forward to receiving a revised version that will be reviewed again by the same referees.

Reviewer 1 ·

Basic reporting

Please see comments below.

Experimental design

Please see comments below.

Validity of the findings

Please see comments below.

Additional comments

This theoretical study addresses an interesting and important question –the mode of development of a secondary sexual trait in relation to its evolution. Unfortunately, it’s not clear to me that this study succeeds in capturing some biologically-important aspects of the problem being modeled, so I’m not sure what to make of the results.

Specific comments:

(1) The relative advantages of age-dependent and age-independent secondary sexual trait expression are likely to depend very much on the relative costs and benefits of “large” trait expression in early life versus the costs and benefits of gradual and continual growth in the trait through the reproductive lifespan. It’s not clear to me that this model captures these costs and benefits. For example, is there a disproportionate viability cost of large trait expression early in the reproductive lifespan (when males are smaller)? The finding that age-independent trait expression tends to evolve at larger trait sizes suggests that this cost is not captured by the simulations. Some other aspects of the simulation and results are also difficult to understand (see below).

(2) The procedure of assigning males with age-independent traits the population-mean trait value at each iteration (L103-114) needs more explanation. I suppose this was done in order to avoid having the competition between the F1 and F2 alleles influenced by differences in mean trait size under these different strategies? My concern is that this peculiar procedure is likely to affect the behaviour of the simulation in ways that make the results difficult to interpret. Moreover, I wonder whether this procedure actually removes some of the biologically meaningful variation. An age-independent allele (as defined in this paper) may be favoured by selection because it allows males to take advantage of mating opportunities in early life, when an age-dependent trait would not yet be developed enough to make the male competitive. By setting the expression of age-independent traits equal to the population mean, do you not lose this effect? I think this procedure needs to be justified in terms of the biological questions of interest in this study, and the consequences of this approach for the behaviour of the simulations should be considered and discussed.

(3) It’s odd that trait and preference were invariably lost when there was any cost to choice. Does this mean that the “good genes” modeled here were of almost no consequence for fitness, or that a genetic correlation between trait and preference couldn’t be established? This result needs more discussion, and perhaps suggests that the simulation parameters need to be adjusted to make the model behave in a more reasonable way. Would it make more sense to start with a model where costly preference can be supported with an age-independent trait, and then ask whether an age-dependent trait allele can invade?

(4) I’m also confused by the conclusion that the evolutionary fate of age-independent traits depends only on their size. What does this mean, biologically? Presumably, a trait with age-independent expression can arise and spread at a small size, but eventually evolve a larger size (i.e., become exaggerated). I think this is how the evolution of such traits is generally envisioned. In the simulations conducted here, trait size cannot evolve, so it’s not clear to me whether the conclusion that trait evolution depends on trait size has biological meaning.

(5) L244-254: Again, I’m confused by some of these statements. Classic LH theory predicts that selection will favour greater investment in reproduction at a young age when mortality rate is high. I’m also not sure what’s meant by “frivolous” traits. I’m also confused by the statement that female mate choice is favoured because it allows them to produce offspring that survive well to old age. Surely, the important parameter here is offspring fitness, not their longevity per se. Subsequent parts of the discussion contain similarly confusing statements (e.g. that males benefit by expressing large traits in old age because selection strength declines with age).

·

Basic reporting

The article is generally well written, aside from the few specific points I make below. It sets out its arguments in a clear and reasonably well-structured manner. There are a few points I would like to make, however, as follows:

P2, line 19-20 it would be nice to have a few references here of examples of some of the “most” sexual selection models to which you refer.

P3, line 47-48: I don’t follow you that males growing age-dependent traits will necessarily rarely encounter choosy females during early reproduction. Do you mean that in species in which age-dependent reproduction occurs, we might expect to observe that males rarely encounter females during early reproduction, since under these circumstances males can afford to “throw away” the few early reproduction opportunities that they get (thrown away since they are unlikely to get a mating with a small ornament)?

P3, line 48-49: Again, is it necessarily the case that most males will have similar trait values at young ages regardless of condition? It could be the case that condition-dependence is strongly reflected throughout the life span, even with age-dependent growth. This could still weaken the reliability of the signal of a large ornament, though, so your point is fundamentally still sound.

P3, line 53: It would be good if you could cite some of the voluminous literature.

P4, line 66: strongly age-dependent traits can increase in frequency in the population – as it stands I initially thought you meant that they could increase in size, either throughout an individual’s lifetime, or on average.

P4, line 79-p5, line 80: I don’t really understand what you mean by “age-0 males supply female allele frequencies”. Surely the female allele frequencies are supplied by the combined efforts of all males, through mating? Do you mean that female allele frequencies are the same as age-0 male allele frequencies?

P5, equation (1): You assume exponential trait growth for the age-dependent trait. I wonder whether you have considered other functions of trait growth? Perhaps a note about why exponential growth is appropriate (i.e. is it actually seen in the wild? Does it simplify the simulation? etc) and what effect other possibilities could have. The note could be here or in the discussion, or both.

P6, line 106: I find it difficult to follow you when it comes to the population mean trait value. First maybe consider changing notation, and using t_bar, or some such symbol, to denote the population mean trait size, and not just t, since as it stands Equation (2) has the index t in the sum, which is potentially confusing. Second, is the population mean trait value averaged over all males, or only F_2-bearing males? Third, I don’t quite follow you as to why the denominator of equation (2) is y¬_max + 1. Finally, does this all mean that age-independent male trait size can in fact change with time from one generation to the next as the population mean value changes? This seems on the face of it somewhat strange, and warrants explanation (though I may have misunderstood).

P7, equation (5): A very small point. I may have missed it, but t_j(y) appears to be undefined – am I correct in thinking that it’s the same as t(j,y), i.e. equation (1) evaluated at C = j? I can see that the rewritten notation makes things much easier here, but I think you should define it.

P7, line 130: “after selection P_i^i” should be “P’_i”?

P7, line 135-136: What is the rationale for the recombination regime you have assumed? Why should condition loci recombine freely, and the other loci be less likely to recombine freely? A word on this would be helpful.

P8, line 141-144: I confess I don’t follow your specification of initial age structure. Is equation (11) the “arbitrary Gaussian survivorship function centred at 0” mentioned on line 142? If so, then you could consider defining it as such in the text. At the moment you start by defining lambda, and then you don’t mention it except in equation (12). Also, why is the left-hand side of equation (12) phi(y)? I thought phi was the female’s mating propensity? Should it be pi(y) here?

P8, line 148: typo, “alelel” should be “allele”.

P9, line 167: perhaps stress fixation versus loss of trait, since the first time I read it through I thought this was fixation versus loss of age-dependence.

P10, lines 169-184: The description here could be simplified if you simply note that strength of selection nu does not seem to affect results in the case of age-independence.

P10, lines 186-192: I think the explanation of Figure 5 could be much clearer. At present it doesn’t address any of the features seen in the figure. I don’t really follow how the trajectories in Figure 5 show the trait declining until condition reaches mutation-selection balance, for example. Is this because there is negative linkage disequilibrium between trait and condition initially in the trajectories, and you know that there must be positive selection on condition? If so then perhaps you could explain this. On a related note, Figure 5 is a bit messy and unclear at present. It’s very hard to tell the difference between many of the six types of point represented on it. Perhaps you should use all squares for “AI” types, and all circles for “AD” types? Though even then, I am at a loss to work out which trajectory belongs to AI Age 1 and AI Age 2, for example. Also it took me a while to work out what “AD” and “AI” are as you haven’t used these abbreviations elsewhere in the document. I think this figure could be a great feature of this work – it’s fantastic to actually see the linkage disequilibrium evolving. At the moment the lack of clarity means the impact of the figure is rather lost, which I think is a shame.

P11, line 195-6: “Fixation of F_2 depended on the associated level of trait expression with initial polymorphism variance at the F locus”. I found this confusing, I’m afraid, because I thought you meant the fixation depending on how much initial polymorphism there was, and then I saw that you had fixed p_F. So maybe you should say that in the case where there was an initial polymorphism, fixated depended on the level of expression, or something like that?

P11, line 200: this relates to my earlier confusion about the mean level of trait expression in the population, but in the case where t(C,y) = t (the mean trait expression), how can the F_2 allele cause males to have smaller average traits than male with age-dependent traits? Surely if t(C,y) = t, the F_2 allele in fact causes males to have the same average trait size as males with age-dependent traits (by definition, in fact)? Maybe I’m missing something?

P11, line 205 Perhaps add “in this region” to the sentence that starts “In other words”, to make it doubly clear that you are only talking about the light grey area of the Figures in question. It’s dead cool that you get polymorphism, by the way!

P11, lines 209-212: “Net selection seems to favor…” I’m not 100% sure I follow this. The ratio of age-independent male trait to young-age age-dependent male trait is b exp{C*y_max} / b exp{C*0} = b exp{C*y_max} / b = exp{C*y_max}, which is independent of the parameters b and nu which affect fixation in Figure 6. Again, maybe I’ve missed something?

P11, line 213: typo, “threwshold”.

P13, lines 262-265: Forgive me, I don’t know the source material, but you say that Proulx et al (2002) find that young males should “downplay their signaling”, and Kokko (1997) finds that young males (of lower condition, admittedly) “should signal more”, and you say these two conclusions are “similar”. Are these findings similar? They seem to me (again, I’m sorry I don’t know the original papers) to be opposed.

P15, lines 302-303: I don’t understand what you mean by “…which could produce strong linkage disequilibrium between condition loci and preference loci under selection against the trait”. Are the preference loci under selection against the trait (if so I don’t understand at all)? Do you mean that, under conditions in which there is selection against the trait, and in which female choice depends on female condition, strong linkage disequilibrium between preference loci and condition loci could be produced? If the latter, is this expected to be positive or negative linkage disequilibrium?

Finally, I wonder whether you could comment in the Discussion about the fact that in your simulation there is a maximum age of 2? What effect, if any, do you think increasing the age limit would have? Why did you choose 2?

Experimental design

No Comments (no experiment).

Validity of the findings

As far as I can tell (see caveats above for the elements of the model and simulation that I don’t follow), the findings are robust and valid. I also think they are very interesting.

I particularly think that a lot of very good (and probably boring to do!) work has gone into sweeping the parameter space, and I commend the author on this.

Additional comments

I think this is a cool piece of work, and deserves to be published, and soon. There are a few small things that I would like cleared up (most of them probably due to my own lack of comprehension rather than any fault with the work), but in general this is a nice study.

---

## Round 0.2 · Minor Revisions

Please make the revised version together with a list of your replies to the comments. The manuscript will be checked by the reviewer again.

·

Basic reporting

The paper is much improved, and I now only have a few relatively minor suggestions.

Chief among these suggestions is that I would like to see acknowledgement of the paper "The Dynamics of Honesty: Modelling the Growth of Costly, Sexually-Selected Ornaments" by Rands et al. from PLoS One in 2011, as well as discussion of how your findings differ from theirs. They also seem to find (in line with the Kokko, and Proulx et al. papers you cite) that at younger ages, high-quality males are likely to have smaller ornaments, while at older ages the reverse is true. They use a stochastic dynamic programming approach rather than your major gene model, so it would be interesting if you would compare the results and approaches, either in the Introduction or the Discussion (or both, as you currently do with the other papers mentioned).

Other points:

Abstract: "Evolution of age-independent traits depends on trait size, whereas evolution of age-dependent traits depends on strength of selection and growth rate (i.e. size)"
This reads like you're saying "Evolution of age-independent traits depends on size, whereas evolution of age-independent traits depends on size". Seems a bit strange. So maybe stress that the former depends ONLY on trait size, while the latter depends on two parameters.

p6, line 110: "mean males of a particular genotype"
Consider amending this to "mean that males..." since the juxtaposition of "mean" and "males" after the parentheses might make the insufficiently careful reader think you are referring to a specific kind of average male, rather than what larger values of b mean.

p7, equation (2): I'm afraid I still don't follow your logic for tbar. Firstly I don't understand the sentence "I calculated tbar at each iteration such that males carrying F2 contributed to the population mean as if their traits were age-dependent". I apologise if I have confused matters further with my comments last time around. I thought that F2 males (in this instance) are simply defined regardless of their age as having an ornament size equal to the mean ornament size for age-dependent males of their condition? But you now seem to be saying that F2 males all have ornament size equal to the mean ornament size of F1 males, regardless of condition. Or maybe not?

Further, I still don't see why there is the denominator of ymax+1. If f(t) is the frequency of males with trait size t, and the sum in the numerator is over all possible discrete trait sizes t, then isn't the numerator already the mean trait size in the population?

p8, line 158: "represents the traits size"
typo, should be "trait size"? Unless you want to adopt "the trait's size" throughout, which I would personally advise against.

p11, line 206: maybe add "initial" before the word "size" in your parenthetical explanation of parameter b?

p13, line 252: "The left-hand column of Figure 5"
Right-hand, surely?

p15, line 300: Maybe point the reader to Figure 1?

p16, line 341-2: "avoid viability selection, by contrast, avoid selection..."
Probably should remove the redundant repetition here.

p17, lines 357-360: I'm afraid I still don't quite get the comparison between Proulx et al and Kokko. Is the conclusion in both papers that poor condition young males should signal more than good condition young males?

p18, line 389: Put the Evans citation in parentheses?

Experimental design

No Comments

Validity of the findings

No Comments

Additional comments

No Comments

---

## Round 0.3 · accepted · Accept

The reviewer is satisfied with the revised manuscrpt. The paper is now suitable for the publication in PeerJ.

·

Basic reporting

No Comments.

Experimental design

No Comments.

Validity of the findings

No Comments.

Additional comments

I am now happy with the manuscript and think it should be published. I commend the author on his patience, candour, and keenness to take suggestions on board.

---

## Author Rebuttal · Round 0.3

Joel J. Adamson
Chapel Hill, North Carolina
Email: adamsonj@ninthfloor.org

Servedio Lab
Biology Department
CB #3280
University of North Carolina
Chapel Hill, NC 27599

Yoh Iwasa
Academic Editor
November 12, 2013

**Subject: Response to PeerJ Submission #2013:08:741:0:0:REVIEW "Evolution of male life histories and age-dependent sexual signals under female choice"**

Dear Editor,

Thank you for considering my manuscript. I am truly grateful for the opportunity to share my work and for the helpful comments of the reviewers. These reviewers have added hugely to the clarity of my manuscript. I have answered the reviewers' comments below to the best of my ability. I greatly appreciate another opportunity for publication with PeerJ.

A note about what follows: I have reproduced my changes here in the rebuttal letter, in pre-2009 fashion. My re-submission will include a diff file indicating all changes to the manuscript file between the original submission and the new one.

My responses are *emphasized* below the corresponding comment, with quotations of the revised text accompanying my responses.

Thank you again for this opportunity. The reviewers' comments were sincerely helpful.

Sincerely,

Joel J. Adamson

- Reviewer 1

**(1)** The relative advantages of age-dependent and age-independent secondary sexual trait expression are likely to depend very much on the relative costs and benefits of "large" trait expression in early life versus the costs and benefits of gradual and continual growth in the trait through the reproductive lifespan. It's not clear to me that this model captures these costs and benefits. For example, is there a disproportionate viability cost of large trait expression early in the reproductive lifespan (when males are smaller)? The finding that age-independent trait expression tends to evolve at larger trait sizes suggests that this cost is not captured by the simulations. Some other aspects of the simulation and results are also difficult to understand (see below).

*I kindly thank the reviewer for pointing out the need for further interpretation. I can see how some of my results are confusing without aiding the reader by using an individual biological perspective. I'm glad to have gone through this exercise and to have improved the current manuscript for my readers. I have added the following paragraph as the fourth paragraph in the Discussion:*

> The failure of the age-independent trait to evolve at lower trait sizes (see Figure 4b) also supports the theory presented above. Using an individual perspective can aid understanding of this result. Consider a population early in the evolution of the trait. The trait and preference occur rarely, and hence viability differences between ornamented and unornamented males dominate the selection differential. Mating success will not form a significant part of fitness due to positive frequency dependence (i.e. rarity of choosy females). From an individual perspective, any trait-bearing male will have lower fitness than an unornamented male. For the trait to be advantageous, it must be large enough for males to gain high mating success despite the rarity of choosy females. On the other hand, for trait-bearing males to avoid viability selection, it must be small. Compare young age-independent males to young age-dependent males in this scenario. Age-independent males that avoid selection never become attractive. Age-dependent males that avoid viability selection, by contrast, avoid selection and become more attractive. Now consider males bearing large traits. Age-independent males that start out with sufficiently large traits immediately attract choosy females, even though they probably die in the next round of selection. If the reader picks a single $b$-value in Figure 4, this represent two potential males: (1) a fully-grown age-dependent male and (2) a young age-independent male. Both males display the same trait size and have the same viability costs. However, the age-dependent male has already lived through two mating episodes and three episodes of selection. After mating both males will die, but the age-dependent male has higher lifetime mating success. Carrying attractive traits at a young age reduces future mating opportunities. We see a threshold of attractiveness in Figure 4b where age-independent males show large enough traits to avoid this cost. Kokko (2001)

has already noted that sexual advertisements are life-history traits. Considering the costs and benefits of advertisements requires evaluation of the entire life cycle and all the organism's interactions and requirements (Badyaev and Qvarnström, 2002).

**(2)** The procedure of assigning males with age-independent traits the population-mean trait value at each iteration (L103-114) needs more explanation. I suppose this was done in order to avoid having the competition between the $F_1$ and $F_2$ alleles influenced by differences in mean trait size under these different strategies? My concern is that this peculiar procedure is likely to affect the behaviour of the simulation in ways that make the results difficult to interpret. Moreover, I wonder whether this procedure actually removes some of the biologically meaningful variation. An age-independent allele (as defined in this paper) may be favoured by selection because it allows males to take advantage of mating opportunities in early life, when an age-dependent trait would not yet be developed enough to make the male competitive. By setting the expression of age-independent traits equal to the population mean, do you not lose this effect? I think this procedure needs to be justified in terms of the biological questions of interest in this study, and the consequences of this approach for the behaviour of the simulations should be considered and discussed.

*I kindly thank the reviewer for this thorough comment, as other reviewers had previously commented on this procedure. This reviewer has offered a solution that encourages me to explain my logic to the reader. The paragraph explaining the procedure now reads:*

> The F locus controls mode of development. Males carrying the $F_1$ allele show age-dependent expression, whereas carriers of the $F_2$ allele express the trait throughout their lives at at one of three levels in a particular simulation: (1) $t(C,0) = b$, the trait value of a 0-year-old; (2) $t(C,y_{max})$, the trait value of the oldest males in the population (still dependent on condition); or (3) $\bar{t}$, the population mean trait value. The third set of simulations sought to create a population where age-independent ($F_2$) males were of intermediate attractiveness, between young or unornamented males and older, age-dependent males. Fixation of age-dependence in this case shows that significant mating advantages accrue later in life, despite a period of lesser attractiveness early in life. Fixation of age-independence, on the other hand, would show that early-life attractiveness and costs were more important than potential gains from mating later in life. Contrast this with the second set of simulations: age-dependent males only reach the attractiveness of their age-independent counterparts, when they reach the final age class ($y_{max}$). Age classes run from 0 (youngest) to $y_{max}$. A youngest age class of 0 conveniently yields young males the trait size of $b$ in Equation (1), as well as using the same indexing convention as the computer simulation. The number of age classes in the population is $y_{max} + 1$.

I calculated $\bar{t}$ at each iteration such that males carrying $F_2$ contributed to the population mean as if their traits were age-dependent:

$$\bar{t} = \frac{\sum_{t=0}^{t_{max}} t f(t)}{y_{max} + 1}$$

where $f(t)$ describes the frequency of males with trait value $t$ over $y_{max} + 1$ age classes. The average is taken over all males, from unornamented males to the maximum trait size of $t_{max} = be^{\mathscr{C} y_{max}}$, where $\mathscr{C}$ represents the largest possible number of condition alleles (i.e. number of condition loci).

I updated $\bar{t}$ in every iteration to ensure that a class of males of intermediate attractiveness persisted in the population. Therefore males carrying $F_2$ received $\bar{t}$ as their trait value for a particular episode of mating, then I updated their values in the next iteration, based on changes in $\bar{t}$. Although this reduces biological realism from an individual perspective, this procedure maintained the population genetic conditions relevant to the question at hand.

**(3)** It's odd that trait and preference were invariably lost when there was any cost to choice. Does this mean that the "good genes" modeled here were of almost no consequence for fitness, or that a genetic correlation between trait and preference couldn't be established? This result needs more discussion, and perhaps suggests that the simulation parameters need to be adjusted to make the model behave in a more reasonable way. Would it make more sense to start with a model where costly preference can be supported with an age-independent trait, and then ask whether an age-dependent trait allele can invade?

*Instead of further explaining this result, as the reviewer suggested, I will respectfully attempt another solution. Results regarding costs of female choice were not crucial to the final analysis, and do not change the manuscript's thesis. I have therefore removed the analysis of female choice costs. This model is poorly suited to modeling costs of choice, but very well-suited to my main topic of interest, the evolution of the male trait. The text now shows an updated female fitness function, with all references to the $s_p$ parameter also removed. I have further clarified in several places within the main text that all female mortality selection results from selection on condition alleles.*

**(4)** I'm also confused by the conclusion that the evolutionary fate of age-independent traits depends only on their size. What does this mean, biologically? Presumably, a trait with age-independent expression can arise and spread at a small size, but eventually evolve a larger size (i.e., become exaggerated). I think this is how the evolution of such traits is generally envisioned. In the simulations conducted here, trait size cannot evolve, so it's not clear to me whether the conclusion that trait evolution depends on trait size has biological meaning.

*I'm glad the reviewer has pointed out this potentially confusing wording of my summary statements. The size referred to is a particular mathematical coefficient, depicted as the horizontal axis of the graphs in the main portion of the results. Given the impor-*

*tance of this parameter (b), I have clarified its introduction and further explained its meaning in the figure captions:   Following the Equation (1) for the male trait:*

> The growth coefficient linearly corresponds to the size of the trait across constant age and condition. Larger values of *b* in a particular population (i.e. simulation) mean males of a particular genotype attain larger trait values than they would in populations with smaller *b*-values.

*Added to the caption for the first figure showing the b-nu plane:*

> Under age-dependence *b* is the value of the trait for a 0-year old male. The *b*-axis corresponds linearly to the trait size of ornamented males ($t(C,y) = be^{Cy}$), such that populations depicted further to the right on the *b*-axis will have larger average trait sizes.

*The offending sentences of the first paragraph in the discussion now read:*

> While an age-dependent trait can evolve by producing small traits at young ages when selection is intense, larger initial trait sizes can only occur under proportionally less intense selection. This contrasts strongly with the age-independent simulations. Size of the trait determines the eventual fixation of the age-independent trait allele, regardless of selection intensity.

**(5) L244-254:** Again, I'm confused by some of these statements. Classic LH theory predicts that selection will favour greater investment in reproduction at a young age when mortality rate is high. I'm also not sure what's meant by "frivolous" traits. I'm also confused by the statement that female mate choice is favoured because it allows them to produce offspring that survive well to old age. Surely, the important parameter here is offspring fitness, not their longevity per se. Subsequent parts of the discussion contain similarly confusing statements (e.g. that males benefit by expressing large traits in old age because selection strength declines with age).

*I thank the reviewer for drawing my attention to the confusing wording of this particular paragraph. The paragraph now reads:*

> Altogether the simulations show that small traits in young males "solve" the dynamical problem of reduced heritability. The results suggest that three factors favor the evolution of age-dependent sexual signals: (1) low adult male mortality; (2) weak selection against the trait; and (3) strong age-dependence, i.e. small initial trait sizes. This result makes sense from a life-history strategy perspective. When adult mortality is lower, young males will invest less in reproductive strategies, producing a negative correlation between expected future reproduction and age-specific investment. Males can afford to invest in mate attraction when adult mortality is low. Age-dependence effectively partitions the life-history of males into two stages dominated by two different fitness components. Viability takes precedence at young ages and mating success becomes more important later in life. This result parallels recent work on an abundant primate showing that juvenile survival followed by mating success form the most crucial fitness components (Courtiol et al., 2012).

- Reviewer 2 (Samuel Tazzyman)

  **P2, line 19-20** it would be nice to have a few references here of examples of some of the "most" sexual selection models to which you refer.

  *I have added citations to several papers making a similar point, to give credit for the idea to Hanna Kokko.*

  **P3, line 47-48:** I don't follow you that males growing age-dependent traits will necessarily rarely encounter choosy females during early reproduction. Do you mean that in species in which age-dependent reproduction occurs, we might expect to observe that males rarely encounter females during early reproduction, since under these circumstances males can afford to "throw away" the few early reproduction opportunities that they get (thrown away since they are unlikely to get a mating with a small ornament)?

  *I have extended this argument to include more details of the evolutionary process I envision. The second point in the list now reads:*

  > Second, frequency dependence crucially affects the origin of costly sexual signals (Kirkpatrick, 1982). If we suppose that both trait and preference arise by mutation, then males growing age-dependent traits will rarely encounter choosy females during early evolutionary stages. Sexual selection will have limited opportunity to increase the trait. Such a trait could be eliminated by selection, or lost to drift in finite populations. This introduces a critical time period the trait must survive before proceeding to fixation.

  **P3, line 48-49:** Again, is it necessarily the case that most males will have similar trait values at young ages regardless of condition? It could be the case that condition-dependence is strongly reflected throughout the life span, even with age-dependent growth. This could still weaken the reliability of the signal of a large ornament, though, so your point is fundamentally still sound.

  *I thank the reviewer for pointing the ambiguity of these statements. The paragraph now reads:*

  > For some kinds of traits, males will have similar trait values at young ages regardless of variation in condition. Age-dependence of traits thereby weakens both the heritability of the trait and the phenotypic correlation between the trait and condition.

  **P3, line 53:** It would be good if you could cite some of the voluminous literature.

  *I thank the reviewer again for pointing out this gap in credit to the authors of the original papers. Instead of citing voluminous amounts of literature, I have cited Hammerstein (1998) which gives a summary of the controversy I cite.*

  **P4, line 66:** strongly age-dependent traits can increase in frequency in the population – as it stands I initially thought you meant that they could increase in size, either throughout an individual's lifetime, or on average.

  *Again I thank the reviewer for pointing out the ambiguity of these sentences. I have added the words "in frequency":*

I show that strongly age-dependent traits can increase in frequency at smaller sizes than age-independent traits, both in the presence and absence of age-independent traits.

**P4, line 79-p5, line 80:** I don't really understand what you mean by "age-0 males supply female allele frequencies". Surely the female allele frequencies are supplied by the combined efforts of all males, through mating? Do you mean that female allele frequencies are the same as age-0 male allele frequencies?

*I have eliminated this confusing parenthetical comment. Since I meant something very different from the reviewer's interpretation, this was a clearly unnecessary and confusing statement. The sentence now reads:*

Each female mates once and lives for one episode of viability selection followed by mating.

**P5, equation (1):** You assume exponential trait growth for the age-dependent trait. I wonder whether you have considered other functions of trait growth? Perhaps a note about why exponential growth is appropriate (i.e. is it actually seen in the wild? Does it simplify the simulation? etc) and what effect other possibilities could have. The note could be here or in the discussion, or both.

*I'm glad the reviewer has pointed out how I can further justify this choice. The discussion contains a note about the choice of trait function and its narrow applicability as a weakness of the model. However, at the reviewer's suggestion I have also added a note about the choice, directly following the presentation of the trait function:*

I chose this exponential function to emphasize three characteristics: (1) all males display the same trait size at age 0, as long as they carry the trait allele; (2) a large disparity in size between young and old males; and (3) simple scaling of the male trait size via the growth coefficient ($b$). Other trait functions do occur in nature (see Johnson and Hixon, 2011; Poissant et al., 2008), and could have different consequences for evolutionary dynamics (see Discussion).

**P6, line 106:** I find it difficult to follow you when it comes to the population mean trait value. First maybe consider changing notation, and using $\bar{t}$, or some such symbol, to denote the population mean trait size, and not just t, since as it stands Equation (2) has the index t in the sum, which is potentially confusing. Second, is the population mean trait value averaged over all males, or only $F_2$-bearing males? Third, I don't quite follow you as to why the denominator of equation (2) is $y_{max} + 1$. Finally, does this all mean that age-independent male trait size can in fact change with time from one generation to the next as the population mean value changes? This seems on the face of it somewhat strange, and warrants explanation (though I may have misunderstood).

*I thank the reviewer for pointing out these points of clarification. I have extensively revised the explanation of the $\bar{t}$ procedures in response to comments by Reviewer #1. Please see my response above for those changes. Additionally, I have added an expla-*

*nation of the number of age classes, in response to Reviewer #2, above the equation for*
$\bar{t}$*:*

> Age classes run from 0 (youngest) to $y_{max}$. A youngest age class of 0 conveniently yields young males the trait size of $b$ in Equation 1, as well as using the same indexing convention as the computer simulation. The number of age classes in the population is $y_{max} + 1$.

**P7, equation (5):** A very small point. I may have missed it, but $t_j(y)$ appears to be undefined – am I correct in thinking that it's the same as $t(j, y)$, i.e. equation (1) evaluated at $C = j$? I can see that the rewritten notation makes things much easier here, but I think you should define it.

*I have added a note after Equation (5) indicating the identity of $t_j$:*

> $\dots$ where $\pi(y)$ represents the frequency of males of age $y$, and $t_j(y)$ represents the traits size of a male of genotype $j$ at age $y$. Condition is not an argument to the $t()$ function in this case, as it was in Equation˜(1) since the genotype $j$ specifies the male's condition.

**P7, line 130:** "after selection $P_i^i$" should be "$P_i'$"?

*I thank the reviewer for pointing out this typo.*

**P7, line 135-136:** What is the rationale for the recombination regime you have assumed? Why should condition loci recombine freely, and the other loci be less likely to recombine freely? A word on this would be helpful.

*I have added a note about the reasons for choosing free versus arbitrary recombination:*

> Condition loci recombine freely ($r = 0.5$) with each other and other loci; other loci recombine at arbitrary frequencies ($0 \leq r \leq 0.5$; see Table 1). Condition loci recombine freely so that they will represent unlinked loci far away in the genome, and could also represent multiple unlinked loci (Rowe and Houle, 1996). The trait and preference loci recombine at arbitrary frequency since prior works show that recombination frequency affects indirect selection on preference (Kirkpatrick and Barton, 1997; Kirkpatrick, 1982).

**P8, line 141-144:** I confess I don't follow your specification of initial age structure. Is equation (11) the "arbitrary Gaussian survivorship function centred at 0" mentioned on line 142? If so, then you could consider defining it as such in the text. At the moment you start by defining lambda, and then you don't mention it except in equation (12).

*I have reworded the sentence, in the hope that my meaning will be clearer:*

> I then used this Gaussian survivorship function centered at 0 to calculate survival probabilities:(followed by Equation (11))

Also, why is the left-hand side of equation (12) $\phi(y)$? I thought $\phi$ was the female's mating propensity? Should it be $\pi(y)$ here?

*I am glad the reviewer has pointed this out in a helpful manner. These errors evaded all my proofreading up to this point.*

**P8, line 148:** typo, "alelel" should be "allele".

*Fixed.*

**P9, line 167:** perhaps stress fixation versus loss of trait, since the first time I read it through I thought this was fixation versus loss of age-dependence.

*I have made the suggested change. The sentence now reads:*

> I analyzed the relative roles of selection intensity and trait size by plotting the equilibrium value of $p_T$ (fixation versus loss of the trait allele) over a plane defined by. . .

**P10, lines 169-184:** The description here could be simplified if you simply note that strength of selection nu does not seem to affect results in the case of age-independence.

*I have attempted to simplify this paragraph according to the Reviewer's suggestions. The second part of the paragraph now reads:*

> The selection parameter $\nu$ determines the pattern of fixation for age-dependent traits, but has no effect on age-independent traits. At $\alpha = 0.6$ the trait fixes in a very small portion of the $b - \nu$ plane near $\nu = 1.0$ in age-dependent simulations. The traits fixes above the threshold $b$ value of 0.20 in the corresponding age-independent simulations. At $\alpha = 1.0$ the regions of fixation are larger, but the qualitative differences between age-dependent and age-independent expression remain: the age-dependent trait fixes in a roughly triangular region characterized by relatively weak selection and containing a region of small initial trait sizes (Figure~4a). This contrasts to age-independent simulations, where again above a threshold size ($b = 0.12$) the trait fixes independently of selection intensity (Figure~4b).

**P10, lines 186-192:** I think the explanation of Figure 5 could be much clearer. At present it doesn't address any of the features seen in the figure. I don't really follow how the trajectories in Figure 5 show the trait declining until condition reaches mutation-selection balance, for example. Is this because there is negative linkage disequilibrium between trait and condition initially in the trajectories, and you know that there must be positive selection on condition? If so then perhaps you could explain this. On a related note, Figure 5 is a bit messy and unclear at present. It's very hard to tell the difference between many of the six types of point represented on it. Perhaps you should use all squares for "AI" types, and all circles for "AD" types? Though even then, I am at a loss to work out which trajectory belongs to AI Age 1 and AI Age 2, for example. Also it took me a while to work out what "AD" and "AI" are as you haven't used these abbreviations elsewhere in the document. I think this figure could be a great feature of this work – it's fantastic to actually see the linkage disequilibrium evolving. At the moment the lack of clarity means the impact of the figure is rather lost, which I think is a shame.

*The figure containing linkage disequilibrium trajectories (Figure 5) has been changed in several ways. I thank the reviewer for pointing out my shortage of explanation and*

*the complexity of the figure. Hopefully I have improved the visual clarity of the figure, despite its inherent complexity.*

1. *The figure now uses color for all six curves*
2. *The Age-dependent curves are found in "warm" colors, and the age-independent curves are displayed in "cool" colors*
3. *The legend clearly spells out the two categories of Age-dependent and Age-independent*
4. *Age 2 is indicated with open boxes to allow viewers to see the points underneath*
5. *The explanation of the figure within the text is more extensive and has a different focus than the paragraph in the prior draft*

Figure 5 shows a sample of linkage disequilibrium trajectories from the four regions defined by fixation and loss of age-dependent and age-independent traits. Each point on a curve represents the statistical association (linkage disequilibrium) between a beneficial condition allele and the ornament allele. A positive association indicates that the trait allele occurs in the same genotype with a beneficial condition allele more often than expected by chance. A negative association indicates that the two alleles are found less often in one genotype than expected by chance. When the alleles randomly associate e.g. at the beginning of the simulation, or when one or both alleles are fixed or lost, the linkage disequilibrium equals zero. Condition alleles do not fix in these simulations due to biased mutation.

These associations indicate the effect of selection on genotypes that act as indicators of condition, i.e. have both a condition allele and a trait allele. When selection favors condition (always) and does not favor the trait, it will drive alleles apart so that they are rarely found in the same genotype, leading to the negative associations found in Figure 5. I caution the reader against interpreting the values in Figure 5 as directly indicative of the strength of overall selection, since the quantities do not include adjustment for age structure. Each point reflects the action of selection on a particular cohort, and not the strength of selection against the trait in the population. For example, the larger absolute values in older age-classes in the figure does not indicate that selection acts more strongly on individuals in that age class. Selection works quite weakly on older age classes, since old individuals are quite rare.

Using these trajectories gives us a slightly clearer picture of the forces acting to generate the pattern of fixation and loss seen in Figure 4. Each panel shows curves for each of three age classes under both age-dependent and age-independent simulations. All four panels show stronger associations in age-independent simulations than in age-dependent simulations. We can also see that magnitude (absolute value) increases from age 0 to age 2. Increasing negative magnitude reflects that high-condition males, likely to survive to older age classes, are less likely to be ornamented. However, this effect is generally much weaker in age-dependent populations than in age-independent

populations (Figure 5a). Males that reach the oldest age class under age-dependence are more likely to be high-condition and ornamented than under age-independence. More raw material for sexual selection remains within a cohort under age-dependence.

The left-hand column of Figure 5 shows trajectories for a larger trait ($b = 0.2$). A similar curve appears for age-dependent traits in Figure 5b as in Figure 5a. However, the age-independent trait responds to selection rapidly, showing a different shape from age-dependent populations (Figure 5c and Figure 5d). Readers should keep in mind when viewing this figure that the age structure of the population, especially under strong selection, biases heavily toward young males. Although linkage disequilibrium increases in older age classes, the associations for age class 0 most closely reflect the overall linkage disequilibrium in the population as a whole.

*I have also added comments in the Discussion:*

Trajectories of linkage disequilibrium support the above theory (Figure 5). Selection finds its greatest power early in the life cycle, dealing with the majority of genetic variation in the largest age class. When the additive component of that variation lies hidden within age-dependent traits, it can pass through to older age classes where it will be fully expressed. As males age they reveal additive variance in condition, thus making them more reliable signalers (as in Proulx et al., 2002). The trajectories illustrate how selection can only eliminate the age-dependent trait when either selection acts strongly or when the young-age trait is large enough. Weak selection (the top row of Figure 5) works even weaker on age-dependent populations, so as to barely change allele frequencies until males get older. Age-dependence weakens selection against the trait.

**P11, line 195-6:** "Fixation of $F_2$ depended on the associated level of trait expression with initial polymorphism variance at the F locus". I found this confusing, I'm afraid, because I thought you meant the fixation depending on how much initial polymorphism there was, and then I saw that you had fixed $p_F$. So maybe you should say that in the case where there was an initial polymorphism, fixated depended on the level of expression, or something like that?

*I have replaced the potentially confusing, long sentence indicated by the reviewer with the following two, more descriptive sentences:*

These simulations began with polymorphism at the F locus, i.e. $F_2$ began at a low, non-zero frequency. Fixation of $F_2$ depended on the level of trait expression chosen for the age-independent males in that simulation, one of (1) $b$; (2) $\bar{t}$; or (3) $t_{max} = b \exp(Cy_{max})$.

**P11, line 200:** this relates to my earlier confusion about the mean level of trait expression in the population, but in the case where t (C,y) = t (the mean trait expression), how can the $F_2$ allele cause males to have smaller average traits than male with age-dependent

traits? Surely if t(C,y) = t, the $F_2$ allele in fact causes males to have the same average trait size as males with age-dependent traits (by definition, in fact)? Maybe I'm missing something?

*I am confident that if the reviewer was missing something, that my confusing wording didn't help him find it. Hopefully my updated explanation of the $\bar{t}$ procedure will clarify my goals in this situation. I have also revised the indicated sentence, so that it now reads:*

> Age-independent males occurred with one particular trait function ($t = b$, $t = \bar{t}$, or $t = b\exp(Cy_{max})$) for a specific simulation: when the $F_2$ allele caused males to have smaller traits than the oldest age-dependent males then $F_2$ was lost everywhere in the $b$-$\nu$-plane.

**P11, line 205** Perhaps add "in this region" to the sentence that starts "In other words", to make it doubly clear that you are only talking about the light grey area of the Figures in question. It's dead cool that you get polymorphism, by the way!

*I have made the suggested change to that particular sentence:*

> In other words, expression remained polymorphic in this region, with the majority of males displaying age-dependence (light gray in Figure 6a).

**P11, lines 209-212:** "Net selection seems to favor…" I'm not 100% sure I follow this. The ratio of age-independent male trait to young-age age-dependent male trait is $bexp\{C * y_{max}\}/bexp\{C * 0\} = bexp\{C * y_{max}\}/b = exp\{C * y_{max}\}$, which is independent of the parameters $b$ and $nu$ which affect fixation in Figure 6. Again, maybe I've missed something?

*I thank the reviewer for indicating his confusion. I have revised the paragraph to better reflect my intended meaning, hopefully ameliorating confusion in the process:*

> When males carrying $F_2$ expressed the trait range of the oldest age-dependent males throughout their lives ($t = b\exp(Cy_{max})$) the $F_2$ allele fixed in an area including the highest intensity of selection (Figure 6). However, the $F_2$ allele also failed to increase in an area of small initial trait values where the trait did fix. In other words, expression remained polymorphic in this region, with the majority of males displaying age-dependence (light gray in Figure 6a).o This effect is even stronger when selection on condition is relaxed ($\mu = 10$; see Figure 6b). The area of trait fixation where F remained polymorphic is larger when $\mu = 10$ and includes regions of more intense selection against the trait as well as requiring larger trait sizes for fixation of $F_2$. Age-dependence occurs at larger trait sizes and under more intense selection when the population displays more variance in condition. This suggests that for a given $b$, age-independence only brings males a mating advantage with strong selection on condition.

**P11, line 213:** typo, "threwshold".

*Fixed.*

**P13, lines 262-265:** Forgive me, I don't know the source material, but you say that Proulx et al. (2002) find that young males should "downplay their signaling", and Kokko (1997) finds that young males (of lower condition, admittedly) "should signal more", and you say these two conclusions are "similar". Are these findings similar? They seem to me (again, I'm sorry I don't know the original papers) to be opposed.

*I thank the reviewer for pointing out these potential ambiguities. I now see that the originally drafted paragraph was not specific enough about Kokko's conclusions. The mentioned portion of the paragraph now reads:*

> Proulx et al. (2002) modeled the situation where male longevity and reproductive opportunities increase — e.g. under a low adult mortality environment — and younger males downplay their signaling relative to lower condition males, preserving resources for survival. Kokko (1997) came to the similar conclusion that young males of lower condition should signal more than their higher-condition cohorts, thus obscuring the observed relationship between genetic quality and trait value. Higher-condition males can afford to downplay signaling more than low-condition males. When males signal in proportion to number of remaining reproductive attempts (a predictor of condition in any age-class), selection favors females that prefer to mate with older males.

**P15, lines 302-303:** I don't understand what you mean by "... which could produce strong linkage disequilibrium between condition loci and preference loci under selection against the trait". Are the preference loci under selection against the trait (if so I don't understand at all)? Do you mean that, under conditions in which there is selection against the trait, and in which female choice depends on female condition, strong linkage disequilibrium between preference loci and condition loci could be produced? If the latter, is this expected to be positive or negative linkage disequilibrium?

*I thank the reviewer for his suggestion. I have added a few words to indicate negative linkage disequilibrium in this evolutionary scenario:*

> Female choice also depends on female condition, which could produce strong negative linkage disequilibrium (i.e. negative correlations) between condition loci and preference loci under selection against the trait. Condition-dependent female choice could therefore broaden the range of parameters where an age-dependent trait can evolve.

**Finally,** I wonder whether you could comment in the Discussion about the fact that in your simulation there is a maximum age of 2? What effect, if any, do you think increasing the age limit would have? Why did you choose 2?

*I kindly thank the reviewer for suggesting that I add something about this. The choice had to do with computer limitations and scientific considerations:*

> I chose the small maximum age of 2 for computational efficiency. Three age classes also yielded young, middle- and old-aged males without producing old males with enormously exaggerated traits. When males had more age classes, old-age traits became very large and far too advantageous to analyze

the tension between sexual selection and viability selection. Males in nature live in more complex age-structured populations with more than three age-classes.